# Analysis of the *Rdr1* gene family in different Rosaceae genomes reveals an origin of an *R*-gene cluster after the split of Rubeae within the Rosoideae subfamily

Ina Menz[1], Deepika Lakhwani[2], Jérémy Clotault[2], Marcus Linde[1], Fabrice Foucher[2], Thomas Debener[1]*

**1** Institute for Plant Genetics, Leibniz Universität Hannover, Hannover, Germany, **2** IRHS, Agrocampus-Ouest, INRA, Université d'Angers, Beaucouzé, France

* debener@genetik.uni-hannover.de

**Data Availability Statement:** All relevant data are within the manuscript and its Supporting Information files.

## Abstract

The *Rdr1* gene confers resistance to black spot in roses and belongs to a large TNL gene family, which is organized in two major clusters at the distal end of chromosome 1. We used the recently available chromosome scale assemblies for the *R. chinensis* 'Old Blush' genome, re-sequencing data for nine rose species and genome data for *Fragaria*, *Rubus*, *Malus* and *Prunus* to identify *Rdr1* homologs from different taxa within Rosaceae. Members of the *Rdr1* gene family are organized into two major clusters in *R. chinensis* and at a syntenic location in the *Fragaria* genome. Phylogenetic analysis indicates that the two clusters existed prior to the split of Rosa and Fragaria and that one cluster has a more recent origin than the other. Genes belonging to cluster 2, such as the functional *Rdr1* gene *muRdr1A*, were subject to a faster evolution than genes from cluster 1. As no *Rdr1* homologs were found in syntenic positions for *Prunus persica*, *Malus* x *domestica* and *Rubus occidentalis*, a translocation of the *Rdr1* clusters to the current positions probably happened after the Rubeae split from other groups within the Rosoideae approximately 70–80 million years ago during the Cretaceous period.

## Introduction

Roses, together with species from the genera *Fragaria*, *Rubus*, *Potentilla*, and *Malus*, belong to the family Rosaceae and are therefore related to economically important fruit crops such as apple and peach [1, 2]. The genus *Rosa*, which includes approximately 200 species, shows a complex evolutionary history due to frequent hybridizations, multiple polyploidizations and recent radiation. The genus is subdivided into four subgenera (*Hulthemia*, *Plathyrhodon*, *Hesperhodos* and *Rosa*) by some authors, whereas others question the subgeneric status of *Hulthemia* and *Plathyrhodon* and propose to include them with the subgenus *Rosa*. The subgenus Rosa itself includes up to 10 sections and approximately 95% of the species [1–5].

**Funding:** The authors received no specific funding for this work. The publication of this article was funded by the Open Access fund of Leibniz Universität Hannover.

**Competing interests:** The authors have declared that no competing interests exist.

Until recently, only fragmented rose genomes were available [6,7]. Recently, two chromo-some scale reference sequences for the diploid *Rosa chinensis* cultivar 'Old Blush' have been published [8,9]. The two sequences were obtained independently from cell cultures regener-ated from microspores and therefore represent haploid segregants of the diploid cultivar 'Old Blush'. The rose genome displays extensive synteny with the *Fragaria vesca* genome with only two major rearrangements [9]. Synteny between *Fragaria* and *Rosa* genes has been observed for TNL genes (TIR (*Drosophila* Toll and mammalian interleukin (IL)-1 receptors), NBS (nucleotide-binding site) and LRR (leucine- rich repeat)) [10].

NBS-LRR genes, which include TNLs and CNLs (CC (coiled-coil)-NBS-LRR), are the larg-est classes of *R*-genes in plants. They are characterized by three domains with different func-tions: the N-termini are thought to be involved in protein-protein interactions, the NBS domain is required for ATP (adenosine triphosphate) binding and hydrolysis, and the LRR-domain is involved in protein-protein interactions and ligand binding [11–14]. NBS-LRR genes have been detected in organisms from green algae to flowering plants and often occur in clusters of related paralogues or as single loci. The number of NBS coding genes in the genome varies widely among different species within the dicots and monocots. Whereas CNLs are found in both monocots and dicots, TNLs occur only in dicots [15]. Among the dicots, the Caricaceae (*Carica papaya*: 54) and Cucurbitaceae (*Cucumis sativus*: 59–71, *C. melo*: 80, *C. lanatus*: 45) families have very low numbers of NBS-encoding genes, whereas the number seems to be greater for some members of Rosaceae (198 NBS genes in *F. vesca* [16] and up to 1303 NBS genes in *Malus* x *domestica* [17,18]). The number of NBS-encoding genes also varies considerably within species, as shown for *Oryza sativa* lines (328–1120 NBS genes) or *Gossy-pium herbaceum* (268–1465 NBS genes) [19]. Different evolutionary dynamics have been pos-tulated, with some clusters comprising fast-evolving genes and others comprising slow-evolving genes [20,14].

In grapevine and poplar, the number of NBS-LRR genes in multi-gene clusters varies between 2 and 10 (mean 4.43) and between 2 and 23 (mean 5.33), respectively [21]. In *Medi-cago truncatula* approximately 50% of NBS domains occur in clusters of at least five genes; the largest cluster (14 genes) occurs on chromosome 6, with a sliding window size of 100 kb. The phylogenetic tree for the 333 non-redundant NBS-LRRs of *M. truncatula* showed that most groups were dominated by sequences from one chromosome and usually from one or a small number of genomic clusters [22]. Molecular characterization of the soybean *Rsv3* resistance locus against multiple soybean mosaic virus strains revealed a cluster of seven highly homolo-gous CNL genes intermixed with 16 other genes in the genotype Williams 82. All seven were also identified in the same order in the genotype Zaoshu 18. The five most likely resistance gene candidates (NBS_A-E) were also sequenced in ten additional soybean cultivars and showed very high sequence similarities [23].

In a *R. multiflora* hybrid (88/124-46), the single dominant TNL gene *Rdr1* (*muRdr1A*), a member of a multigene family of at least nine highly similar clustered TNLs (*muRdr1A-muR-dr1I*), confers broad-spectrum resistance against black spot [24,25]. The sizes of all TNLs for the *Rdr1* locus, except *muRdr1D* (interrupted by 6957-bp transposable-element insertion within intron), range from 4085 to 5920 bp with sequence similarities between 78.0% and 99.5%. The domain structure of typical TNL proteins is reflected by the following intron-exon structure: the first exon contains the TIR domain, the second exon contains the NBS domain, and the fourth (or in case of TNL–*muRdr1I*, the third exon) contains an LRR domain [25].

A region from *R. rugosa* (subsection *Cinnamomeae*), homologous to the *Rdr1* locus in *R. multiflora* (subsection *Synstylae*), was identified with a high degree of synteny that included some flanking non-TNL genes coding for a yellow stripe-like protein, ubiquitin and a TOPLESS-RELATED protein [10]. An analysis of 20 TIR-NBS-LRR (TNL) genes obtained

from *R. rugosa* and *R. multiflora* revealed illegitimate recombination, gene conversion, unequal crossing over, indels, point mutations and transposable elements as mechanisms of diversification. Additionally, an orthologous locus in *F. vesca* (strawberry) was identified that contains a homologous TNL gene family and the flanking genes. In contrast, in *Prunus persica* (peach) and *Malus* x *domestica* (apple), only the flanking genes can be detected in syntenic positions, and the genes homologous to the *Rdr1* family are distributed on two different chromosomes. Phylogenetic analysis of TNL genes from five *Rosaceae* species showed that most of the genes occur in single species clades, indicating that recent TNL gene diversification began prior to the split of the Rosoideae (*Rosa*, *Fragaria*) from the Spiraeoideae (*Malus*, *Prunus*) [26,18] however, not considering genomic positions of individual genes in much detail.

With the availability of chromosome scale assemblies of the *R. chinensis* 'Old Blush' genome, we were interested in analysing the full complexity of the *Rdr1* gene family at the genomic level including the number of paralogues and their position in the genome. Furthermore, we tried to elucidate the advent of this TNL family in the Rosaceae in respect of its emergence in the taxonomic lineage leading to the genus *Rosa* and its evolutionary dynamics after taxonomic separation of *Fragaria* and *Rosa* as well as within the genus *Rosa*. For this we used data from different taxonomic levels, including re-sequencing data for nine rose species recently published along the with the 'Old Blush' genome sequences.

## Results

### *Rdr1* homologs in 'Old Blush' and *F. vesca*

The screening of the haploid genomes derived from 'Old Blush' for TNLs homologous to *Rdr1* from *R. multiflora* resulted in seven complete TNLs for HapOB1 (OB1-A-G) and 21 for HapOB2 (OB2-A-U). A comparative analysis of TNLs from HapOB1 and HapOB2 showed that the following are identical: OB1-B and OB2-D, OB1-C and OB2-I and OB1-D and OB2-G. The sequences of all *Rdr1* homologs are listed in S1 Dataset.

Phylogenetic analysis of the TNLs from *R. multiflora*, HapOB1 and HapOB2 using the maximum likelihood method resulted in the tree shown in S1 Fig. The phylogram shows that a group of three TNLs (OB1-G, OB2-T, OB2-U) are clearly separated from all other TNLs. OB1-G is located on chromosome 5, and OB2-T and OB2-U are located on chromosome 2. All other TNLs from HapOB1 and HapOB2 are located on chromosome 1 and are clustered in two distinct groups (1 and 2) that are highly supported by a bootstrap value of 100%. TNLs from the *R. multiflora Rdr1* cluster are clustered in group 2. The genomic organization of HapOB1- and HapOB2-TNLs on chromosome 1 is shown in Fig 1. For HapOB2, all but three (OB2-A, -B, -I) of the 16 complete TNLs are organized in two clusters at the distal end of the chromosome. Cluster 1 (with a size of 76 kb) contains 37 protein-coding genes, of which 28 displayed significant similarities to entries in the GenBank database, including six complete TNL genes (OB1-C to OB1-H) and some truncated TNL genes (three TIR-domains, one LRR-domain and two NBS-LRR genes). Cluster 2 (with a size of 163 kb) contains 28 protein coding genes, of which 23 displayed significant similarities to entries in the GenBank database, including ten TNL genes (OB1-J to OB1-S) and one additional LRR-domain. TNLs from HapOB1 are also organized in two clusters at the distal end of chromosome 1. Gene prediction identified three TNLs (OB1-B through OB1-D) for cluster 1 and two TNLs (OB1-E and OB1-F) for cluster 2. Additionally, two TIR-domains, two NBS-LRR genes and one LRR-domain could be found within the cluster.

To determine the reasons for the unusually small number of *Rdr1*-TNLs at the two cluster positions in the HapOB1 genome, we analysed the DNA from haploid tissue that had been used for sequencing of the HapOB2 genome as well as DNA from the original diploid OB cultivar with the Rd1LRR microsatellite marker from the LRR region of the gene family. Seven of

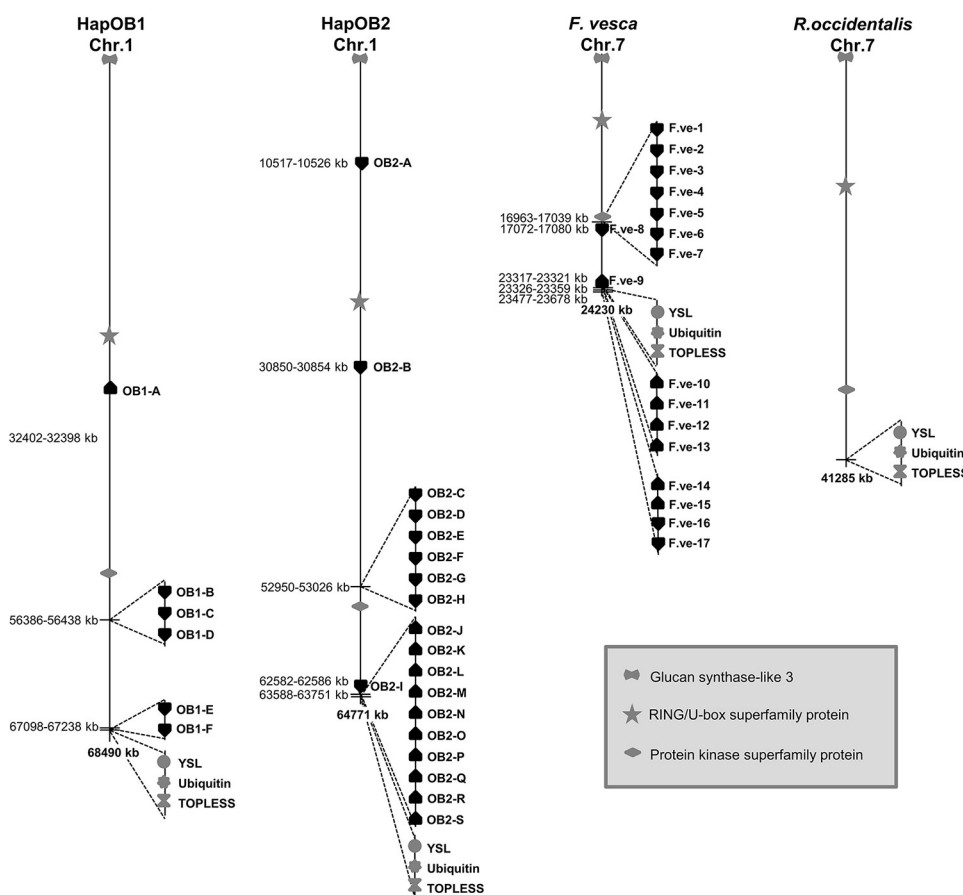

**Fig 1. Genomic organization for *Rdr1* homologs in HapOB1, HapOB2, *Fragaria* and *Rubus*.** Shown are: chromosome 1 of HapOB1 and HapOB2 with the upper cluster 1 (OB1-B through OB1-D; OB2-C through OB2-H) and the lower cluster 2 (OB1-E and OB1-F; OB2-J through OB2-S); chromosome 7 of *F. vesca* with cluster 1 (F.ve-1 through F.ve-8) and cluster 2 (F.ve-9 through F.ve-17); and chromosome 7 of *Rubus occidentalis* (no *Rdr1* homologs found). Positions of three syntenic genes (glucan synthase-like 3, RING/U-box superfamily protein, protein kinase superfamily protein) and the *Rdr1* flanking genes YSL (yellow-stripe-like protein), Ubiquitin and TOPLESS-RELATED protein are shown in grey.

the 19 genes of HapOB2 contained perfect primer binding sites and were detected on high resolution polyacrylamide gels, whereas 21 fragments were detected in DNA of the diploid OB (S2 Fig). The small number of *Rdr1* genes in the HapOB1 genome are likely to be an artefact, possibly due to a problem with the assembly or resulted from recombination events prior to the isolation of the independent haploid callus line from microspores. Therefore, the HapOB1 sequence was not considered in further analyses.

The genomic organization of the TNLs on the chromosome in the two clusters corresponds to the two groups formed in the phylogenetic tree shown in S1 Fig. OB2-C through OB2-H are clustered in group 1, whereas OB2-J through OB2-S are clustered in group 2.

Analysis of the genes surrounding the clusters revealed a high level of synteny between HapOB1, HapOB2 and *F. vesca* (S1 Table).

The separation of the clusters in two different groups in the phylogenetic tree is further supported by a number of diagnostic sites in the derived amino acid sequences. At two positions (90 and 166), sequences of groups 1 and 2 display unique differences. At three additional positions (348, 688 and 868), one of the two groups displays unique amino acids that are replaced by two or more different sites in the other group.

In addition, the nucleotide diversity differs within each group. Though averages within the group total nucleotide differences are similar for both groups (327 for group 1 and 339 for group 2), the ratio of non-synonymous to synonymous sites is higher in group 2 (2.75) than in group 1 (1.81).

In addition to the TNLs from HapOB1 and HapOB2, the *F. vesca* genome was screened for *Rdr1* homologs. A total of 19 *Rdr1* homologs were found in *F. vesca*, of which 17 are located on chromosome 7 and two are located on chromosomes 1 and 2 (F.ve-19 and F.ve-18, respectively). The 17 TNLs on chromosome 7 are organized in clusters at the distal end of the chromosome: cluster 1 contains seven TNLs, and clusters 2 and 3 contain four TNLs (Fig 1). Phylogenetic analysis of the TNLs from HapOB1, HapOB2 and *F.* vesca shows that the rose genes for the two clusters from chromosome 1 are grouped with the *Fragaria* genes from the clusters on chromosome 7, whereas the genes located on other chromosomes are clearly separated from this group (Fig 2). Chromosome 1 of rose is syntenic with chromosome 7 of *Fragaria* [9]. Furthermore, the rose genes in cluster 2 form a group (group 2) with *Fragaria* genes in cluster 2 and 3, and each of them build a distinct single species clades within this group. In contrast, the genes from cluster 1 do not form strictly single species clades within group 1, but one clade with mixed species and two single species clades.

## TNL structure in other Rosaceae

In a previous study [10], no *Rdr1* homologs could be observed in *P. persica* and *M. domestica* genomes at syntenic positions. Updated genome assemblies have been released since then, and these might have been corrected for assembly errors around repeat regions. We therefore analysed the genomes again for the presence of *Rdr1* homologs at syntenic positions. Rose chromosome 1 (where *Rdr1* is located) presents a good synteny with chromosome 2 in peach and chromosomes 1, 2 and 7 in apple [9]. Nevertheless, no homologous sequences for the *Rdr1* gene were found at these positions, confirming the previous results.

In addition, we also analysed syntenic positions in *Rubus occidentalis*, a species from the Rosoideae sub-family, for which a chromosome scale assembly recently became available [27]. Synteny analysis of the genes surrounding the TNL clusters revealed no *Rdr1* homologs in syntenic positions for *P. persica*, *M.* x *domestica* and *R. occidentalis* (S1 Table). In *Prunus* and *Malus*, more distantly related *Rdr1* homologs were only detected in non-syntenic positions (S3 Fig), whereas in a draft genome from *Potentilla micrantha*, another species from the Rosoideae, several contigs contained *Rdr1* homologs. The genes P.mi-12 and -13 are located on contig 1260 together with genes coding for a yellow stripe-like protein, ubiquitin and a TOPLESS-RELATED protein flanking the *Rdr1* locus in *R. multiflora* and *R. rugosa*, indicating that *Rdr1* homologs are present at syntenic positions in *P. micrantha*. Analysis for *Rdr1* homologs identified 19 for *F. vesca*, three for *R. occidentalis*, 10 for *Malus* x *domestica*, 17 for *P. persica* and 11 for *P. micrantha* (S2 Table).

Phylogenetic analysis showed that the non-syntenic *Rdr1* homologs from *P. persica*, *M.* x *domestica* and *R. occidentalis* are clearly separated from *Rdr1* homologs of OB and *F. vesca*, which are located on chromosome 1 (OB) and 7 (*F. vesca*) (Fig 3). Furthermore, some of the *P. micrantha Rdr1* homologs are grouped together with the TNLs from OB and *F. vesca*, which are located on chromosome 1 (OB) and 7 (*F. vesca*) consistent with clusters of these genes in syntenic positions for the *Rdr1* clusters.

## Rdr1 homologs from other rose species

Analysis of seven additional recently available genome sequences [8] identified 15 *Rdr1* homologs for *R. damascena*, three for *R. persica*, eight for *R. moschata*, 13 for *R. xanthina spontanea*,

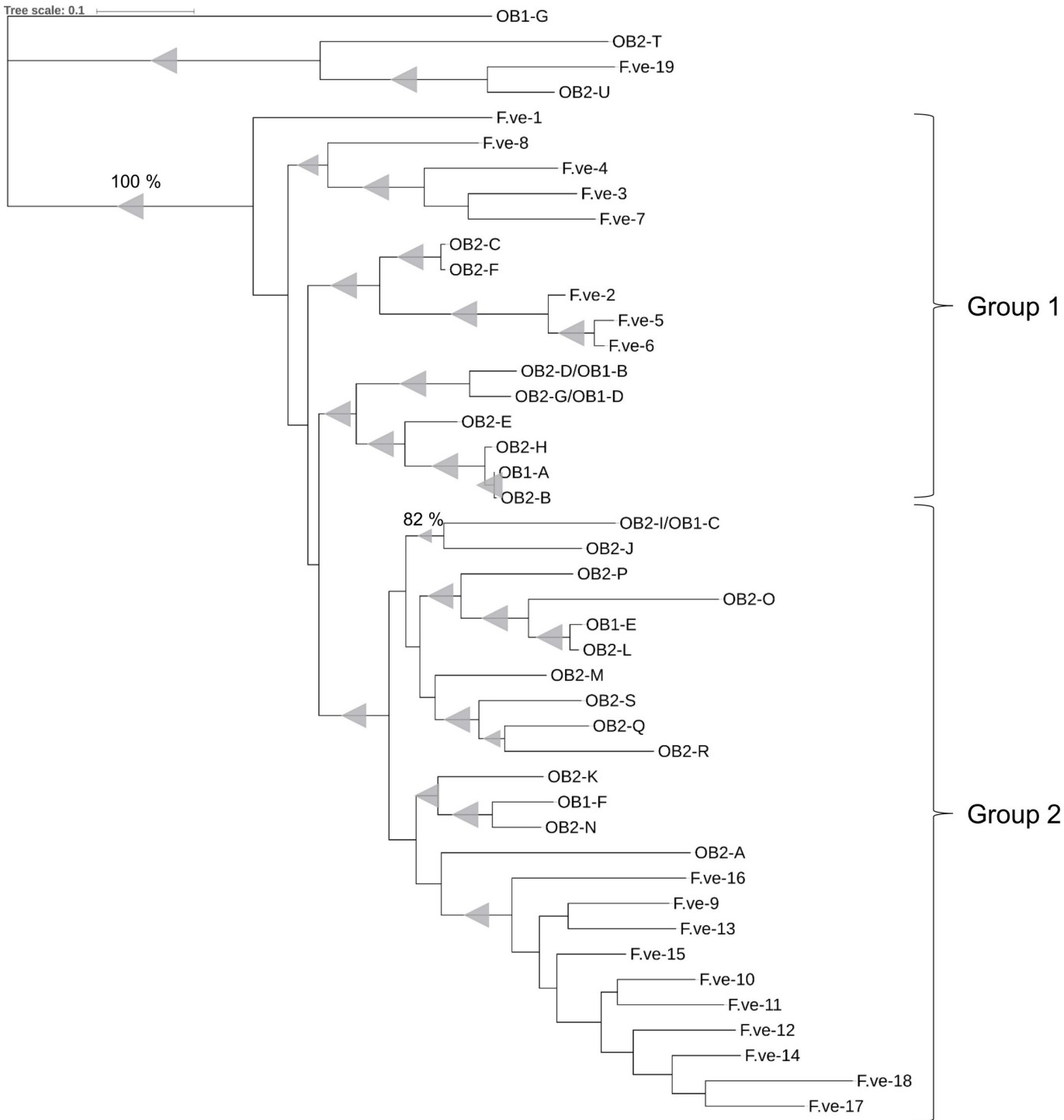

**Fig 2. Phylogenetic analysis of the amino acid sequence of HapOB1-, HapOB2- and *F. vesca*-TNLs homologous to *Rdr1* in *R. multiflora*.** The Maximum Likelihood method based on the JTT matrix-based model was used to calculate the phylogenetic tree. Test of phylogeny was performed using the bootstrap method with 500 replicates. Branches reproduced in less than 75% of bootstrap replicates are collapsed. Bootstrap values are indicated as triangles, whereas the smallest value represents 82% and the largest 100%.

13 for *R. chinensis var. spontanea*, nine for *R. laevigata* and 12 for *R. minutifolia alba* (Table 1). Until recently, only highly fragmented genomes have been available for these rose species, which makes a chromosomal classification for TNLs homologs in *Rdr1* difficult.

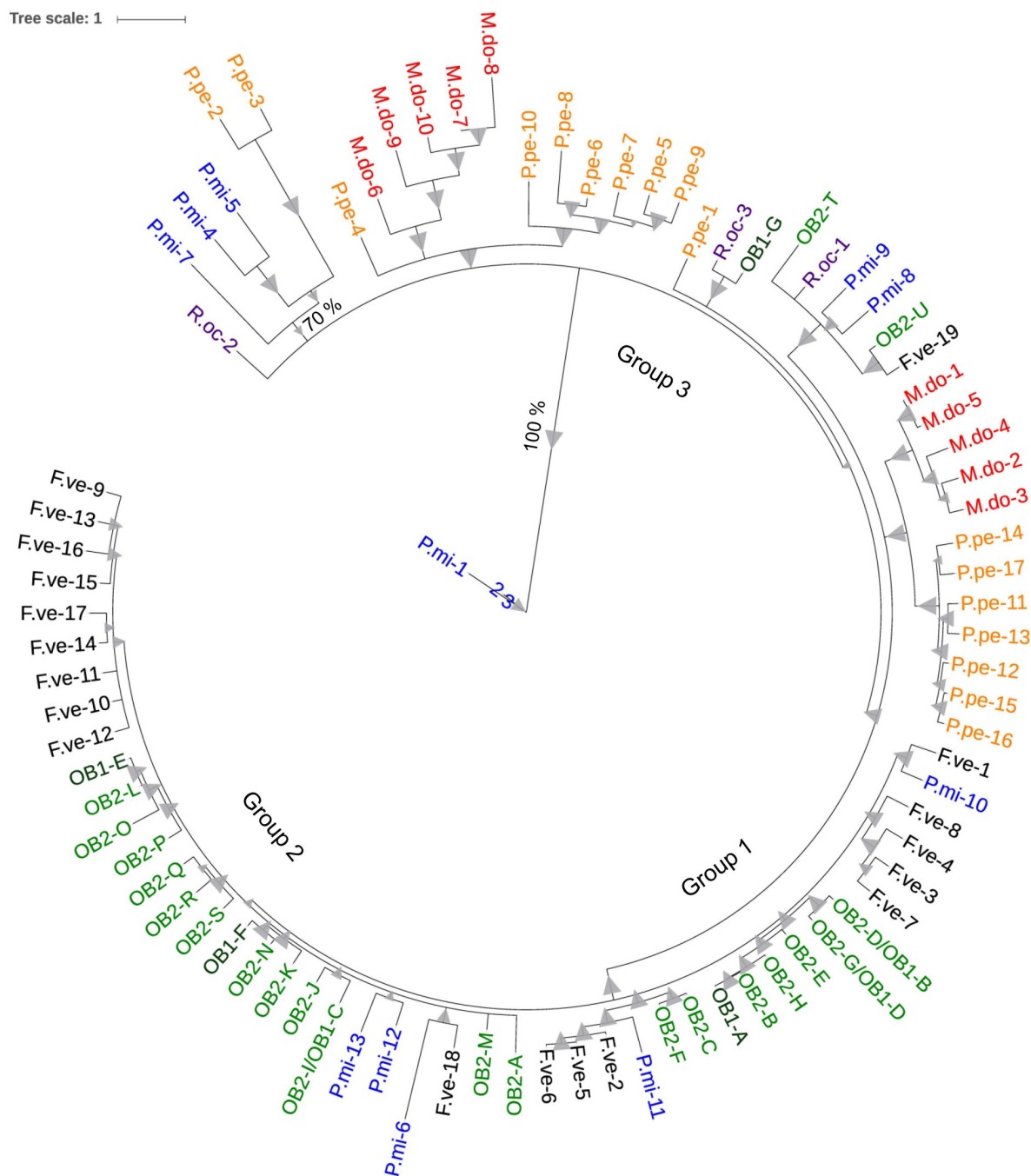

**Fig 3. Phylogenetic analysis of the amino acid sequence of TNLs from different Rosaceae family members homologous to Rdr1 of R. multiflora.** The Maximum Likelihood method based on the JTT matrix-based model was used to calculate the phylogenetic tree. Test of phylogeny was performed using the bootstrap method with 500 replicates. Branches reproduced in less than 60% of bootstrap replicates are collapsed. Bootstrap values are indicated as triangles, whereas the smallest value represents 70%, and the largest value represents 100%. For a better visualization, Rdr1 homologs for the different Rosaceae family members are coloured as follows: HapOB1/2 (OB1+2: dark green), M. domestica (M.do: red), F. vesca (F.ve: black), P. persica (P.pe: orange), P. micrantha (dark blue), R. occidentalis (purple). The protein alignments are shown in S2 Dataset.

**Table 1. List of *Rdr1* homologs found in different rose species.**

| Species | Abbreviation | TNLs |
|---|---|---|
| *R. multiflora* (from[25]) | R.mu (A-I) | 9 |
| *R. rugosa* (from [10]) | R.ru (A-K) | 11 |
| **HapOB1** | OB1 (A-G) | 7 |
| **HapOB2** | OB2 (A-U) | 21 |
| *Rosa damascena* | R.da (1–15) | 15 |
| *Rosa persica* | R.pe (1–3) | 3 |
| *Rosa moschata* | R.mo (1–8) | 8 |
| *Rosa xanthina spontanea* | R.xa (1–13) | 13 |
| *Rosa chinensis* var. *spontanea* | R.ch (1–13) | 13 |
| *Rosa laevigata* | R.la (1–9) | 9 |
| *Rosa minutifolia alba* | R.mi (1–12) | 12 |

Based on the observation that *Fragaria Rdr1* homologs from syntenic clusters form phylogenetic groups with rose homologs for *Rdr1*, we computed a phylogenetic tree to identify homologs from other rose species (Fig 4). For *R. multiflora* and *R. rugosa* TNLs already obtained by [25] were used. The most conspicuous group (group 3), with high bootstrap support, contains single TNLs from HapOB1/2 and *Fragaria* located on different chromosomes outside the two syntenic clusters. They are grouped together with two *R. chinensis*, two *R. minutifolia*, two *R. moschata* genes and one *R. xanthina* gene, which also most likely represent genes from outside the syntenic clusters. All *Rdr1* homologs of HapOB, *R. multiflora* [25], *R. rugosa* [10] and *Fragaria*, known to derive from cluster 2, fall into one highly supported large group (group 2) that also includes sequences from all other rose species.

Within group 2, *Rdr1* homologs from *Fragaria* form a distinct sub-group, whereas most of the other rose sequences form mixed sub-groups with no clear single species clades. In contrast, sequences clustered in group 1 do not form genus-specific sub-groups, but *Fragaria* and rose sequences form mixed sub-groups.

## Discussion

More than 50% of the NBS-encoding genes are organized in clusters in the genome for many species such as *Arabidopsis* (64–71%), rice (50–74%), potato (73%), *Medicago* (80%) and apple (80%) [17,28]. Furthermore, these clusters are not evenly distributed between chromosomal positions. In *Medicago truncatula*, chromosome 6 contains approximately 34% of all TNLs, and chromosome 3 harbours approximately 40% of all CNLs [22]. In apple, approximately 56% of all identified RGAs are located on six of the 17 chromosomes, with 25% on chromosome 2 alone; whereas in grapevine, 80% of TNLs were located on chromosomes 5, 12 and 18 [28,21]. In tomato, the majority of NBS-LRRs are located close to the telomeres, where recombination occurs frequently, while few were detected in regions called "cold spots" for recombination [29]. An accumulation of RGAs in sub-telomeric regions was also described for apple [28].

Previously, we characterized members of the *Rdr1* gene family, among which the *Rdr1* gene confers resistance to black spot [24,25] and forms a cluster of closely related genes. As no complete genome was available at that time, our analyses were constricted to the region captured by BAC contigs and previous versions of the *Fragaria* genome (and others). This research used the high-quality chromosome-scale assembly of the OB genome to analyse the structure of this gene family in more detail. Recently, two high-quality sequences at the chromosome scale from two independent haploids from the same cultivar 'Old Blush' were obtained [8,9].

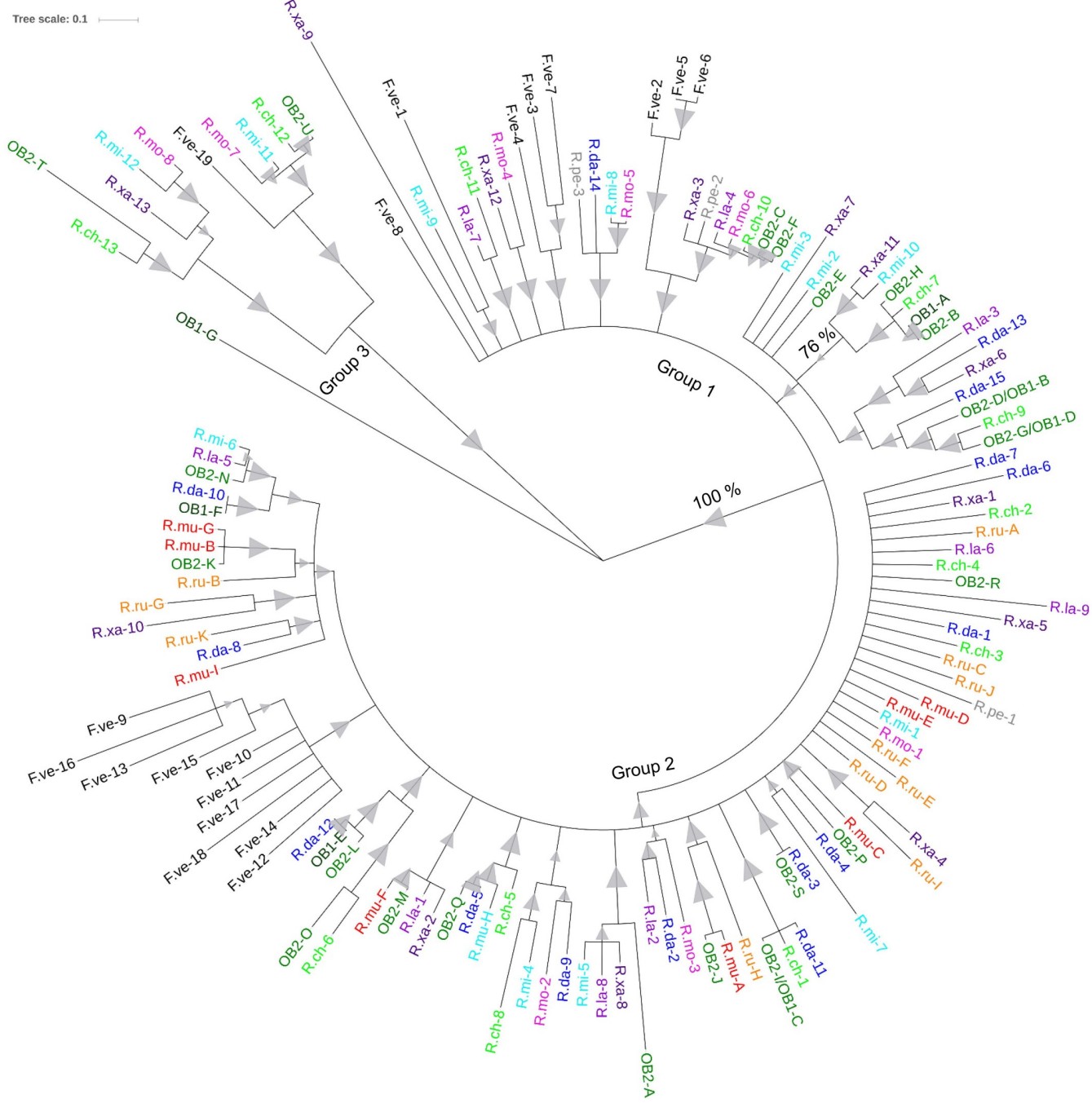

**Fig 4. Phylogenetic analysis of the amino acid sequence of *Rdr1* homologs from different rose species and *Fragaria*.** The Maximum Likelihood method based on the JTT matrix-based model was used to calculate the phylogenetic tree. Test of phylogeny was performed using the bootstrap method with 500 replicates. Branches reproduced in less than 75% of bootstrap replicates are collapsed. Bootstrap values are indicated as triangles, whereas the smallest value represents 76%, and the largest value represents 100%. For better visualization, *Rdr1* homologs for the different species are coloured as follows: HapOB1/2 (OB1/2: dark green), *R. multiflora* (R.mu: red), *F. vesca* (F.ve: black), *R. rugosa* (R.ru: orange), *R. damascena* (R.da: dark blue), *R. persica* (P.pe: grey), *R. moschata* (R.mo: pink), *R. xanthina* (R.xa: dark purple), *R. chinensis* (R.ch: neon green), *R. laevigata* (R.la: purple), *R. minutifolia* (R.mi: light blue). The protein alignments are shown in S3 Dataset.

However, even a high-quality assembly might contain assembly errors around regions of highly similar paralogues for large gene families. Evidence for this is provided by our analysis

of the HapOB1 genome [8], which only predicts seven *Rdr1* paralogues at the chromosome 1 positions in contrast to the situation in the HapOB2 genome [9], where 21 TNLs were annotated. Our access to source DNA was restricted to the original 'Old Blush' diploid genotype and the haploid material used to generate the HapOB2 genome; therefore, we can only state that the total number of amplified copies of the *Rdr1* paralogues from the original diploid is twice as high as that from the HapOB2 genome (S2 Fig). Thus, the fact that the HapOB1 genome contains only seven paralogues might either be due to assembly errors or recombination events prior to the isolation of the independent haploid callus line from microspores in which meiosis had already occurred. Both processes could result in the elimination of some copies of the *Rdr1* family members in the genome sequence. However, this remains unclear because only a fraction of the *Rdr1* paralogues can be amplified with our primer combination and we do not have access to the HapOB1 DNA to analyse this DNA directly.

Our analysis shows that two major clusters and two single genes are located on chromosome 1 and further relatives are located on chromosome 2 of OB. Phylogenetic analysis shows that the two major clusters form different groups, which indicates an independent development of the two clusters. Related sequences found on chromosome 2 are clearly distinct from those on chromosome 1 and are therefore not treated as members of the same family.

Re-analysis of the *Fragaria* genome reveals a similar structure with TNL clusters at syntenic positions. A phylogram of complete *Rdr1* sequences for the *Fragaria* and OB genomes show that *Fragaria* group 2 and rose group 2 are closer to each other than to *Fragaria* group 1 and rose group 1. Furthermore, genes from group 1 only form mixed groups with single species clades, whereas the genes from group 2 form single species clades. As both clusters were present before the taxa emerged, the likely cause is a faster evolution within group 2. This could be due to the known processes by which R-genes evolve (including higher rates of recombination, gene conversion and birth and death processes), which led to a concerted evolution of genes in group 1. A similar observation has been made for inbred lines of maize, in which some paralogues are organized in genotype-specific subgroups [30]. A possible reason for the larger dynamics of group 2 in both Rosa and Fragaria might be the telomeric position in relation to group 1 a factor known to promote higher rates of recombination.

A re-analysis of the latest versions of the apple and peach genomes confirmed earlier results [10] that there are no *Rdr1*-like TNL clusters at syntenic positions in these genomes. The former conclusion remains that the emergence of the *Rdr1* clusters must have formed after the Amygdaloideae split from the Rosoideae. A high-quality genome of *R. occidentalis* recently became available; therefore, we also checked for the presence of our cluster in *Rubus*, which was not present at a syntenic position.

Genome information for *P. micrantha*, identifies a larger number of fragments, which shows that there are 5 contigs with *Rdr1* homologs.

One of these contigs (contig no. 1260) contains two *Rdr1* homologs and conserved genes flanking group 1 in roses [10]. This indicates that *Rdr1* homologs in *Potentilla* are in a putative syntenic position to the group 1 cluster in roses.

The other genes fall into groups of OB sequences that are in both clusters as well as on chromosome 2 in roses. This agrees with the Rosaceae phylogeny which places *Potentilla* and *Fragaria* into sister groups of the Potentilleae within the Rosoideae. The timeline for the evolution of the Rosaceae [26] led us to conclude that the *Rdr1* cluster was translocated to its current position after the Rubeae split from other groups within the Rosoideae approximately 70–80 million years ago during the Cretaceous period.

A larger phylogram, including 137 sequences from ten species of *Rosa*, shows that all rose sequences form mixed clusters with few exceptions. Therefore, single species clades for the rose genes within group 1 have not been developed yet. Not all rose species can be easily

differentiated taxonomically, and most are highly interfertile; this underlines a close relationship between these taxa and may be one reason for the lack of differentiation of group 1 genes.

This study is a first step in the analysis of the evolution of genes from the *Rdr1* family in roses. However, we must keep in mind that assembly processing for clustered duplicated genes can lead to assembly errors. We can then hypothesize that some genes which were studied could represent consensus sequences for several real existing genes. As shown with HapOB2, an assembly obtained from long reads should result in a high-quality chromosome scale assembly for these regions. However, the lower than expected number of *Rdr1* homologs in the HapOB1 assembly, developed from PacBio reads, shows that this might only be a general principle.

## Material and methods

### Origin of sequences

For *R. multiflora* (HQ455834.1) and *R. rugosa* (JQ791545), previously published contigs spanning the *Rdr1* locus were used [10,25]. The genomes of 'Old Blush', HapOB1 [8] and *R. damascena* Mill. were downloaded from NCBI (https://www.ncbi.nlm.nih.gov/), whereas the haploid genome of 'Old Blush', HapOB2 [9] was downloaded from a genome browser (https://iris.angers.inra.fr/obh/). The whole genomes of *F. vesca*, *Malus x domestica*, *P. persica*, *R. occidentalis* and *P. micrantha* were downloaded from the Genome Database for Rosaceae (https://www.rosaceae.org/).

Additionally, sequences of the rose species *R. persica*, *R. moschata*, *R. xanthina spontanea*, *R. chinensis* var. *spontanea*, *R. laevigata* and *R. minutifolia alba* were used ([9], assemblies unpublished). The origins of all used sequences are listed in Table 2.

### Analysis of the Rd1LRR microsatellite marker in 'Old Blush'

The *Rdr1*-TNLs in the 'Old Blush' genome were amplified from DNA for the haploid tissue that had been used for sequencing the HapOB2 genome as well as from DNA of the original diploid OB cultivar using the Rd1LRR microsatellite marker, presented in the coding

**Table 2. Origin of sequences used in this study.**

| Species | Reference | Information |
|---|---|---|
| *R. multiflora* | [25] | HQ455834.1 |
| R. rugosa | [10] | JQ791545 |
| **HapOB1** | [8] | NC_037088.1-NC_037094.1 |
| HapOB2 | [9] | PRJNA445774 |
| *R. damascena* | Unpublished | LYNE00000000.1 |
| F. vesca | [31] | v4.0.a1 |
| *M. x domestica* | [32] | GDDH13 v1.1 |
| P. persica | [33,34] | v2.0.a1 |
| *R. occidentalis* | [27] | v3.0 |
| P. micrantha | [35] | v1.0 |
| *R. persica* | [9] | SRP143586 |
| R. moschata | [9] | SRP143586 |
| *R. xanthina spontanea* | [9] | SRP143586 |
| R. chinensis var. spontanea | [9] | SRP143586 |
| *R. laevigata* | [9] | SRP143586 |
| R. minutifolia alba | [9] | SRP143586 |

sequences for the NBS-LRR members, and analysed on a LiCor 4300 DNA-analyser as previously described [3].

## Gene prediction and annotation

Regions homologous to the *Rdr1* locus were identified for all species using local BLAST searches implemented in Bioedit [36]. The BLASTn method was conducted with the *muRdr1A*-sequence as a query and an E-value of 1.0E-20.

Gene prediction and annotation was performed using FGENESH and AUGUSTUS (http://www.softberry.com; http://augustus.gobics.de/). The protein domains were determined using PfamScan ([37], https://www.ebi.ac.uk/Tools/pfa/pfamscan/). Only genes with a size larger than 2 kb and coding for all three protein domains (TIR, NB-ARC, LRR) were used for further phylogenetic analyses.

**Sequence alignment and construction of phylogenetic trees.** The predicted amino acid sequences of the *Rdr1* homologs of *R. multiflora*, *R. rugosa*, *F. vesca*, HapOB1, HapOB2, *R. damascena*, *R. chinensis* var. *spontanea*, *R. laevigata*, *R. minutifolia alba*, *R. persica*, *R. moschata* and *R. xanthina spontanea* were aligned in MEGAX using MUSCLE (Multiple sequence comparison by log- expectation, [38]) with default options.

For the aligned *Rdr1* homologs from the different species, phylogenetic trees were constructed in MEGAX [39] using the maximum likelihood (ML) method with the Jones-Taylor-Thornton matrix-based model using a discrete gamma distribution with empirical frequencies (JTT+G+F) [40]. The best model was estimated using MEGAX. Initial trees for the heuristic search were obtained automatically. The tree topology was tested via a bootstrap analysis with 500 replicates. For a better visualization of the phylogenetic trees the software Tree Of Life (iTOL) version 4.2.3 [41] (https://itol.embl.de/) was used. Nucleotide diversity within groups of sequences was computed in MEGAX using nucleotide differences among aligned sequences.

The analysis of synonymous and non-synonymous sites was performed in MEGAX by aligning the amino acid sequences of sets of coding DNA-sequences and analysing the DNA differences with the Nei-Gobojori model [42] for 1314 positions in the final dataset.

## Synteny analysis

For the synteny analysis of the two clusters, genes surrounding the clusters were selected based on the rose reference sequence [9]. Reciprocal BLAST were performed against the most recent available Rosaceae genomes: *Fragaria vesca* [31, 45], *Prunus persica* [34], *Malus domestica* [32] and *Rubus occidentalis* [43]. The order of the homologous genes was checked on the genome browser of the GDR website (https://www.rosaceae.org/tools/jbrowse, [44]).

## Supporting information

**S1 Fig. Phylogenetic analysis of the amino acid sequence for *R. multiflora Rdr1*-TNLs and homologous TNLs of HapOB1 and HapOB2.** The maximum likelihood method based on the JTT matrix-based model was used to calculate the phylogenetic tree. A test of phylogeny was performed using the bootstrap method with 500 replicates. Branches reproduced in less than 75% of bootstrap replicates are collapsed. Bootstrap values are indicated as triangles, whereas the smallest value represents 87% and the largest 100%.
(TIF)

**S2 Fig. Results from Rd1LRR microsatellite PCR.** DNA from haploid tissue that had been used for sequencing the OB2 genome as well as DNA of the original diploid OB cultivar was used in a PCR with Rd1LRR microsatellite primers (Terefe and Debener, 2011). PCR products

were separated on a 6% polyacrylamide gel.
(TIF)

**S3 Fig. Genomic organizations of TNLs homologous to *Rdr1* from *Prunus persica* and *Malus* x *domestica*.**
(TIF)

**S1 Table. Results of micro-synteny analysis outside the *Rdr1* family clusters.**
(XLSX)

**S2 Table. Positions and annotation of TNLs homologous to *Rdr1* on the different chromosomes of Old Blush (OB1+2), *F. vesca* (F.ve), *Prunus persica* (P.pe), *Malus domestica* (M. do), *Rubus occidentalis* (R.oc) and *Potentilla micrantha* (P.mi).**
(DOCX)

**S1 Dataset. Coding sequences of all used genes in this study.**
(TXT)

**S2 Dataset. Muscle alignment of protein sequences used for the phylogram shown in Fig 3.**
(TXT)

**S3 Dataset. Muscle alignment of protein sequences used for the phylogram shown in Fig 4.**
(TXT)

## Author Contributions

**Conceptualization:** Ina Menz, Jérémy Clotault, Marcus Linde, Fabrice Foucher, Thomas Debener.

**Data curation:** Ina Menz, Fabrice Foucher, Thomas Debener.

**Formal analysis:** Ina Menz, Deepika Lakhwani, Fabrice Foucher, Thomas Debener.

**Investigation:** Ina Menz, Deepika Lakhwani, Jérémy Clotault, Marcus Linde, Fabrice Foucher, Thomas Debener.

**Methodology:** Ina Menz, Deepika Lakhwani, Jérémy Clotault, Marcus Linde, Fabrice Foucher, Thomas Debener.

**Project administration:** Fabrice Foucher, Thomas Debener.

**Resources:** Ina Menz, Deepika Lakhwani, Jérémy Clotault, Marcus Linde, Thomas Debener.

**Supervision:** Fabrice Foucher, Thomas Debener.

**Validation:** Ina Menz, Deepika Lakhwani, Jérémy Clotault, Marcus Linde, Fabrice Foucher, Thomas Debener.

**Visualization:** Ina Menz, Jérémy Clotault, Fabrice Foucher, Thomas Debener.

**Writing – original draft:** Ina Menz, Jérémy Clotault, Marcus Linde, Fabrice Foucher, Thomas Debener.

**Writing – review & editing:** Ina Menz, Jérémy Clotault, Marcus Linde, Fabrice Foucher, Thomas Debener.

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
