## [Decision Letter · Decision Letter 0]

18 Nov 2019

PONE-D-19-26378

Analysis of the Rdr1 gene family in different Rosaceae genomes reveals an origin of an R-gene cluster after the split of Rubeae within the Rosoideae subfamily

PLOS ONE

Dear Dr. Linde,

Thank you for submitting your manuscript to PLOS ONE. After careful consideration, we feel that it has merit but does not fully meet PLOS ONE’s publication criteria as it currently stands. Therefore, we invite you to submit a revised version of the manuscript that addresses the points raised during the review process.

We would appreciate receiving your revised manuscript by Jan 02 2020 11:59PM. To enhance the reproducibility of your results, we recommend that if applicable you deposit your laboratory protocols in protocols.io, where a protocol can be assigned its own identifier (DOI) such that it can be cited independently in the future. For instructions see: http://journals.plos.org/plosone/s/submission-guidelines#loc-laboratory-protocols

We look forward to receiving your revised manuscript.

Kind regards,

Eric Jellen, Ph.D.

Academic Editor

PLOS ONE

Journal Requirements:

"We thank the Leibniz Universität Hannover and the INRA (Université d’Angers, SFR

419 4207 QuaSaV) for supporting this study, as well as the ‘Région Pays de la Loire’ for

420 the resequencing of eight wild species (Genorose project in the framework of RFI

421 ‘Objectif Végétal’).".

 "The authors received no specific funding for this work.".

Additional Editor Comments (if provided):

Reviewers agree the manuscript is technically sound but dispute its suitability for PLOS ONE. Editor has agreed it merits the publication criteria pending minor revisions, mainly those indicated by reviewer #1, for example clarifying methodology for PCR markers, grammatical errors, etc.

Reviewers' comments:

Reviewer's Responses to Questions

**Comments to the Author**

1. Is the manuscript technically sound, and do the data support the conclusions?

Reviewer #1: Yes

Reviewer #2: Partly

2. Has the statistical analysis been performed appropriately and rigorously? 

Reviewer #1: Yes

Reviewer #2: Yes

3. Have the authors made all data underlying the findings in their manuscript fully available?

Reviewer #1: Yes

Reviewer #2: No

4. Is the manuscript presented in an intelligible fashion and written in standard English?

Reviewer #1: Yes

Reviewer #2: Yes

5. Review Comments to the Author

Reviewer #1: This study reveals some of the evolutionary history of the extremely economically valuable Rdr1 gene family to the horticultural industry, which has been shown to confer Black Spot resistance. Their phylogenetic analysis revealed that there are two major clusters of these genes within Rosa Old Bush, and Fragaria vesca. Genes from this family tend to cluster in regions of the chromosome that are more prone to evolve.

Their findings are of value in elucidating the evolutionary history of the Rdr1 genes in Rosaceae. Understanding their location will also assist in future breeding efforts of crops within the family, which could potentially lead to improved disease resistance. Their findings reflect those of this gene family in other genera that have been previously published.

The research conducted and the conclusions made from the results are supportive and appropriate. The authors discuss the differences between hapOB1 and hapOB2 with regards to the discrepancy in the number of Rdr1 genes. There are many Rdr1 genes missing from the genome of hapOB1 that were also not successfully PCR amplified. The authors suggest that the Rdr1 genes in hapOB1 are not likely to be an artefact and were not discovered because of poor sequencing. This is not entirely convincing though, especially since the use of PacBio technology was utilized for the generation of the hapOB1 sequence data and PCR amplification was not successful for the loci. Is it possible that a deletion event(s) occurred during meiosis that was necessary for the development of the haploid line, especially since these genes are near the end of the chromosome? The data suggest that this portion of the chromosome has somehow been lost. Sequencing errors from PacBio are possible, but to also fail at being able to amplify the genes suggest that a biological event may have occurred at that point in the genome.

There are no methods describing how PCR markers were used to amplify the genes in the Old Bush genomes, nor was there sequence information of the primers themselves. Also, the authors should discuss (at least in the introduction) how and where hapOB1 and hapOB2 were generated and why DNA is not available from hapOB1. Also, if hapOB1 DNA is not available, how did the authors perform PCR to screen for the Rdr1 genes in the first place? Are these haploid lines available? If so, from where?

More work could be done to clarify the introduction to the paper. It provides many details about TNL genes across the plant kingdom (which probably could be scaled back some) but doesn’t give the paper much direction as far as what specific questions the authors were trying to answer during their research or how their findings will fill in an important knowledge gap in Rosaceae. As stated before, they could discuss the origins of hapOB1/2 in the intro to make their work more understandable by non-specialists.

There are a couple of minor grammatical errors to note:

Line 60 change “between” to “among”

Line 84 change “an” to “a”

Reviewer #2: The authors describe an effort to clarify the evolutionary history of an important gene family in Rosaceae. Leveraging new genomic resources they collected sequences for a genomic region that has clusters of TNL genes at conserved locations in Rosaceae, specifically focusing on Rose. The evolutionary history of this gene family in specific lineages of Rosaceae seems to be the main message, however the broader context and relevance for either the field of evolutionary genomics or Rosaceous horticulture is not clear.

The introduction is a carefully curated collection of facts about Rose phylogeny (it's complicated) and TNLs in plants, even going as far as delving into gene intron/exon structure and patterns and patterns of genome structure in other plant lineages. Then this section abruptly ends with a declaration that the authors were interested in looking more closely at this gene family in Rose. It's not at all clear what this work helps us understand - I can think of some reasons, especially with regard to understanding the role of these important R-genes in Rose, and how this information could be very valuable to breeders, or how understanding the dynamics of the changes seen in this gene family can help us learn about how genomes evolve. This information, the potential impact of the work, or clear hypotheses, are lacking from the intro, which leaves the reader asking "so what?" from the very beginning.

Technically I think the work is sound, though admittedly my background is not strong in methods of building phylogenies. I recommend that someone with a strong background in phylogenetics/phylogenomics weigh in on the appropriateness of methods. I would think that more bootstrap reps were needed for these challenging phylogenies, and there needs to be a better way of indicating bootstrap support at critical tree positions besides triangles that vary slightly in size.

The context that would make this work appealing to a broad audience (i.e. that of PONE) is lacking. The conclusions drawn are appropriate (especially if another reviewer can vet the methods), but why it's important for anyone except a narrow audience is not clear, and even then, these potential narrow audiences are not addressed. The authors could write to perhaps ornamental horticulturalists, or Rose breeders, or Rose taxonomists - as it is now it's not written for the audience of PLOSONE.

The written english is technically good, the figures are OK, yet the organization and narrative quality need work. The figures are referenced out of order, which causes confusion. Table S3 is first, and it's not a Table, it' a data file as are Tables S4 and S5 - these are alignments, they should be provided as data files. Table S1 has an empty tab with labeled Tabelle3.

Overall I think the work is sound, but the presentation needs a lot of work. Critically, the authors would need to develop a narrative with broad appeal for PONE. If they decide to target a more narrow audience, they still need to work on developing a clear message and context. The authors themselves say this is a first small step, and mainly confirm prior results and persistant questions in the discussion.

6. PLOS authors have the option to publish the peer review history of their article (what does this mean?). If published, this will include your full peer review and any attached files.

Reviewer #1: No

Reviewer #2: No

---

## [Author Response · Author response to Decision Letter 0]

16 Dec 2019

Dear Reviewers,

thank you very much for the valuable comments on our manuscript. We made the suggested changes in the manuscript and the supplemental files to improve our publication.

---

## [Editor Report · Decision Letter 1]

19 Dec 2019

Analysis of the Rdr1 gene family in different Rosaceae genomes reveals an origin of an R-gene cluster after the split of Rubeae within the Rosoideae subfamily

PONE-D-19-26378R1

Dear Dr. Linde,

We are pleased to inform you that your manuscript has been judged scientifically suitable for publication and will be formally accepted for publication once it complies with all outstanding technical requirements.

With kind regards,

Eric Jellen, Ph.D.

Academic Editor

PLOS ONE
---

## [Editor Report · Acceptance letter]

8 Jan 2020

PONE-D-19-26378R1 

Analysis of the Rdr1 gene family in different Rosaceae genomes reveals an origin of an R-gene cluster after the split of Rubeae within the Rosoideae subfamily 

Dear Dr. Linde:

I am pleased to inform you that your manuscript has been deemed suitable for publication in PLOS ONE. Congratulations! Your manuscript is now with our production department. 

With kind regards,

on behalf of

Dr. Eric Jellen 

Academic Editor

PLOS ONE